# Tradeoffs between resources and risks shape the responses of a large carnivore to human disturbance

Kirby L. Mills [1✉], Jerrold L. Belant[2], Maya Beukes[3], Egil Dröge [4,5], Kristoffer T. Everatt[6,7,8], Robert Fyumagwa[9], David S. Green[10], Matt W. Hayward[11,12,13], Kay E. Holekamp [14,15], F. G. T. Radloff [16], Göran Spong[17], Justin P. Suraci [18], Leanne K. Van der Weyde[19,20], Christopher C. Wilmers[21], Neil H. Carter [22,23] & Nathan J. Sanders[1,23]

Wide-ranging carnivores experience tradeoffs between dynamic resource availabilities and heterogeneous risks from humans, with consequences for their ecological function and conservation outcomes. Yet, research investigating these tradeoffs across large carnivore distributions is rare. We assessed how resource availability and anthropogenic risks influence the strength of lion (*Panthera leo*) responses to disturbance using data from 31 sites across lions' contemporary range. Lions avoided human disturbance at over two-thirds of sites, though their responses varied depending on site-level characteristics. Lions were more likely to exploit human-dominated landscapes where resources were limited, indicating that resource limitation can outweigh anthropogenic risks and might exacerbate human-carnivore conflict. Lions also avoided human impacts by increasing their nocturnal activity more often at sites with higher production of cattle. The combined effects of expanding human impacts and environmental change threaten to simultaneously downgrade the ecological function of carnivores and intensify human-carnivore conflicts, escalating extinction risks for many species.

[1] Department of Ecology and Evolutionary Biology, University of Michigan, Ann Arbor, MI, USA. [2] Department of Fisheries and Wildlife, Michigan State University, East Lansing, MI, USA. [3] Senckenberg Research Institute and Nature Museum, Terrestrial Zoology, Frankfurt, Germany. [4] WildCRU, Department of Biology, University of Oxford, Tubney, UK. [5] Zambian Carnivore Programme, Mfuwe, Zambia. [6] Panthera, New York, NY, USA. [7] Centre for African Conservation Ecology, Nelson Mandela University, Port Elizabeth, South Africa. [8] Greater Limpopo Carnivore Programme, Limpopo, Mozambique. [9] Wildlife Conservation Initiative, Arusha, United Republic of Tanzania. [10] Institute for Natural Resources, Portland State University, Portland, OR, USA. [11] Conservation Science Research Group, School of Environmental and Life Science, University of Newcastle, Callaghan, NSW, Australia. [12] Centre for African Conservation Ecology, Nelson Mandela University, Qgeberha, South Africa. [13] Centre for Wildlife Management, University of Pretoria, Tshwane, South Africa. [14] Department of Integrative Biology, Michigan State University, East Lansing, MI, USA. [15] Program in Ecology, Evolution, and Behavior, Michigan State University, East Lansing, Michigan, MI, USA. [16] Department of Conservation and Marine Sciences, Faculty of Applied Sciences, Cape Peninsula University of Technology, Cape Town, South Africa. [17] Molecular Ecology Group, SLU, 901 83, UMEÅ, Sweden. [18] Conservation Science Partners, Inc., Truckee, CA, USA. [19] Cheetah Conservation Botswana, Gaborone, Botswana. [20] San Diego Zoo Institute for Conservation Research, Escondido, CA, USA. [21] Environmental Studies Department, University of California, Santa Cruz, CA, USA. [22] School for Environment and Sustainability, University of Michigan, Ann Arbor, MI, USA. [23] These authors jointly supervised this work: Neil H. Carter, Nathan J. Sanders. ✉email: kimills@umich.edu

Humans and wildlife are increasingly sharing the world's landscapes and interacting at an unprecedented scale[1-3]. Wildlife responses in these shared landscapes determine both their survival and their roles in the evolving social-ecological systems that govern human and wildlife livelihoods[4-7]. Ongoing global environmental changes and growing human pressures (e.g., expanding agricultural lands) are likely to further intensify wildlife responses to humans. Altered wildlife and human distributions, behaviors, and interactions have the potential to escalate human-wildlife conflicts and increase extinction risks for many species[8-12]. Understanding the complexities of the spatio-temporal responses of wildlife to human disturbance is a key first step in identifying the conditions necessary to foster socio-ecological coexistence between humans and wildlife[13].

In human-dominated landscapes, wildlife must balance the tradeoffs between resource acquisition and the potentially lethal risks of human encounters[14-16]. This is particularly true for large carnivores when they target livestock as prey because threats to livestock often prompt retaliatory killing of large carnivores, a leading cause of large carnivore decline worldwide[16,17]. The flexible spatiotemporal behaviors of large carnivores allow them to hunt prey while reducing harmful interactions with other predators or humans[18-22]. Some large carnivores exploit this flexibility to alter their activity patterns and space use in response to risks from humans, although these changes may come with fitness consequences or restructure ecological processes[23,24]. For example, many carnivore species avoid human encounters by being active mostly at night, referred to as temporal avoidance[25,26], which can influence interspecific competition, predator-prey dynamics, and ecosystem function[18,27,28]. Carnivore species can also avoid human-dominated areas altogether via spatial avoidance, which effectively limits their available habitat and can increase competition, contribute to heightened extirpation risks, restructure community dynamics, and reduce biodiversity[17,29-31]. Conversely, some species exploit resource subsidies provided by humans by accessing human-dominated areas at low-risk times, but often resulting in heightened human-carnivore conflicts over food resources[32,33].

Heterogeneity in risk-averse behaviors among species is well-documented[34-36], but it is unclear how widespread these behaviors are within a given carnivore's distribution. We also lack empirical insights on the mechanisms shaping these behaviors across gradients of anthropogenic disturbance and ecological conditions. For example, resource scarcity could prompt predators to expand their realized spatial or temporal niches to meet their metabolic needs at the expense of their safety[37-40]. Varying intensity and types of human disturbance within a species' range – such as spatially static infrastructure and land use versus temporally dynamic human presence – can also moderate risk-averse behaviors, even leading to the habituation of wildlife to humans in some cases[41,42]. Range-wide syntheses of large carnivore responses to disturbance are therefore needed to better understand their ecology and conservation in shared, complex landscapes[43].

We investigated the spatiotemporal responses of lions (*Panthera leo*) to human disturbances across their range. Almost half of the current range of lions lies outside of protected area boundaries, requiring lions to regularly navigate degraded human-dominated landscapes[44-46]. Accelerating human population growth across Africa (particularly near protected areas) and expanding agricultural lands will intensify the human-lion interface and potential conflicts due to livestock depredation[47,48]. The additional stressors of ongoing prey depletion[49,50] and climate changes[51,52] are expected to exert unprecedented pressures on lions and other wildlife in coming years. Although lions have been extensively studied, large-scale patterns in lion responses to humans remain unclear and are clouded by heterogeneous results among lion populations.

To investigate how lions navigate shared landscapes, we conducted a meta-analysis that examines lion spatiotemporal responses to human disturbance over gradients of disturbance intensity and resource availability. By synthesizing the impacts of human disturbance on large carnivore niche space, we improve our ability to predict how continued human development and global environmental change will impact wildlife behaviors and ultimately their survival. Specifically, we examined whether 1) lions avoid the risks of human disturbance on average across sites by exhibiting lower space use and more nocturnal activity in areas of high disturbance (i.e., reducing spatiotemporal overlap with disturbance); 2) lions exhibit stronger human avoidance behaviors at sites with higher overall human disturbance, including livestock production; and 3) lion avoidance of human disturbance is mitigated by variable primary productivity that could limit wild prey availability.

Through a systematic literature review, we compiled lion occurrence and activity data from 23 studies, representing 31 independent study sites that span 40% of the contemporary range of lions[53] from expansive protected areas to subsistence agricultural lands (Fig. 1). We assessed the strength of lion responses to human disturbance at each site by comparing the intensity of space use and/or the proportion of nocturnal activity between treatments of low and high human disturbance. We quantified effect sizes for lion responses to disturbance using the standardized mean difference (SMD) for spatial responses and the log-response ratio (RR) for temporal responses. We used meta-analytic mixed-effects models to calculate the average effect size for lion responses across studies, weighted by study variance ($SMD_w$ and $RR_w$), as well as investigate the impacts of site-level anthropogenic and environmental conditions on the strength of lion responses to disturbance. We extracted the average and spatial variation of the Human Footprint Index[54] (HFI) at each study site to represent a comparable measure of the intensity and spatial heterogeneity in human disturbance across sites. We also compared lion responses to disturbance with the intensity of livestock production at each site[55]. To assess the effects of resource availability on lion avoidance of disturbance, we used satellite-derived measures of vegetation greenness (i.e.,

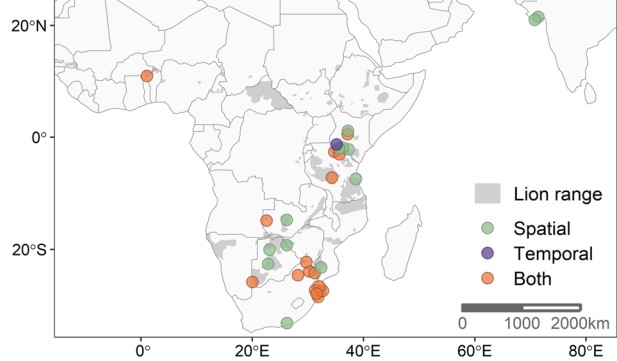

**Fig. 1 The geographic distribution of study sites (n = 31) included in the meta-analysis across the current extant range of lions (estimated by the IUCN).** Colors of points indicate which type of response was calculated for each study site (spatial only [green], temporal only [purple], or both [orange]). Points are not precisely representative of study area centroids to reduce overlap of nearby points (e.g., the positions of two points for studies in Gir National Park, India, are slightly adjusted so that both are visible). Map lines delineate study areas and do not necessarily depict accepted national boundaries. Citation for IUCN range map layer: *Panthera and WCS 2016. Panthera leo. The IUCN Red List of Threatened Species. Version 2022-2.*

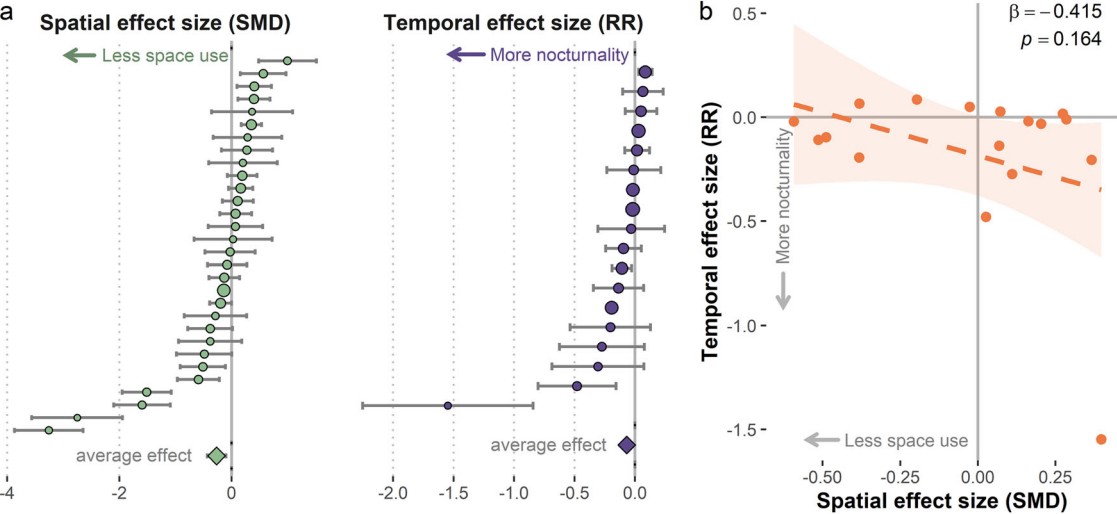

**Fig. 2 The effects of human disturbance on lion space use and nocturnality across all 31 study sites.** Negative effect sizes indicate lion avoidance of high human disturbance domains at a given study site (e.g., lower space use or higher levels of nocturnal activity). **a** Forest plots of lion spatial responses (SMD = standardized mean difference, n = 30) and temporal responses (RR = log response ratio, n = 18) among all studies with 95% CI. Studies are organized in descending order of effect size, and point sizes are proportional to the inverse variance used to weigh each study. Diamond shaped points depict the average weighted effect ($SMD_W$ and $RR_W$) of both response types. **b** The relationship between spatial and temporal effect sizes for study sites where both responses could be extracted (n = 17). The outlier point in the bottom right corner of the plot is discussed in the "Results" section.

Normalized Difference Vegetation Index [NDVI]), including the overall average NDVI during the study period as well as spatial and temporal variability in NDVI. NDVI correlates directly with primary productivity and thus provides a measure of primary resource availability, such as surface water and habitat that strongly influence lion foraging strategies[56–58]. NDVI also generally predicts wild prey abundances across Africa, though this relationship can be decoupled when wild prey populations in productive sites are decimated by overharvesting or other anthropogenic impacts[59,60]. Characteristics of the studies used in the meta-analysis (such as the lion observation method, human disturbance measures, or study area size) were also tested for their influence on observed lion responses to ensure that heterogeneity in study design across sites did not unduly influence our results.

## Results

**Avoidance of human disturbance**. We identified 31 total study sites from 492 search results, resulting in 30 estimates of spatial responses and 18 estimates of temporal responses by lions to humans (Fig. 1). Lions tended to avoid within-site human disturbance with lower space use ($SMD_w = -0.268$, 95%CI ± 0.165) and a 7.08% (95%CI: 3.34, 10.94%) increase in nocturnal activity in high disturbance areas of a given site ($RR_w = -0.068$, 95% CI ± 0.035) (Fig. 2a, Table 1). There was high heterogeneity in effect sizes for spatial responses among studies ($\tau^2$ 95%CI: 0.108, 0.449; $I^2$ 95%CI: 82.2, 95.0; Table 1), and somewhat lower heterogeneity (as well as lower confidence in heterogeneity estimates) in lion temporal responses among studies ($\tau^2$ 95%CI: 0.006, 0.251; $I^2$ 95%CI: 66.7, 98.7; Table 1). We extracted both spatial and temporal effects of human disturbance on lion activity for 17 study sites. Across those sites, there was no relationship between the magnitude of spatial and temporal responses ($\beta = -0.415$ [SE 0.28], $R^2 = 0.12$, $F = 2.138$, $df = 15$, $p = 0.16$; Fig. 2b).

**Ecological and anthropogenic conditions**. The best mixed-effects models (Table 1) revealed substantial effects of NDVI patterns on the strength of lion responses to human disturbance within each site. Lions were more likely to reduce spatial overlap

with high disturbance areas in sites where there was high spatial variation in site-level NDVI ($NDVI_{sp}$: $\beta = -0.893$, 95%CI ± 0.221; Fig. 3b), as well as more consistent NDVI over time ($NDVI_{tm}$: $\beta = 0.218$, 95%CI ± 0.184; Fig. 3e). Primary productivity may also influence the strength of lion temporal responses, as we observed stronger nocturnal shifts at sites with high average NDVI ($NDVI_a$: $\beta = -0.015$, 95%CI ± 0.013; Fig. 3d) and high spatial variation in NDVI ($\beta = -0.061$, 95%CI ± 0.016; Fig. 3b). As expected, monthly NDVI and precipitation were positively correlated at 28 of 29 African study sites included in the meta-analysis (Fig. S3, Table S4), confirming that ecosystem productivity was in part dictated by climate trends and usually increased during rainy months.

Site-level variation in human pressure also influenced how lions responded to human disturbance within each site. Lions avoided high disturbance areas both spatially and temporally where there was low spatial variation in overall human pressure ($HFI_{sp}$: $\beta_{spatial} = 0.286$, 95%CI ± 0.194; $\beta_{temporal} = 0.061$, 95%CI ± 0.022; Fig. 3a). Lions also exhibited shifts towards more nocturnal behavior in response to human disturbance at sites with high levels of cattle production ($\beta = -0.102$, 95%CI ± 0.058; Fig. 3c). However, tests for residual heterogeneity in both the spatial ($Q_E = 143.99$, $df = 26$, $p < 0.001$) and temporal models ($Q_E = 38.53$, $df = 13$, $p = 0.002$) suggested that additional unexplored variables may influence the strength of lion responses.

**Study characteristics**. The responses of lions to direct human activity versus infrastructure did not differ (spatial responses: $F = 0.72$, $df = 2$, $p = 0.49$; temporal responses: $F = 1.54$, $df = 2$, $p = 0.25$; Fig. S4). Similarly, none of the other assessed study characteristics (i.e., lion observation method, study area size, study season, study duration, and median study date) influenced the observed effect sizes for the spatial responses of lions (Figs. S4 and S5). In contrast, the strength of the shifts in lion nocturnal activity appeared to change over time and depended on the observation method used. Lions were less likely to use (or researchers less likely to observe) increased nocturnality to avoid humans in studies conducted in later years ($F = 20.31$, $df = 16$, $p < 0.001$; Fig. S5). Stronger shifts towards nocturnal activity were

**Table 1 Top performing mixed-effects models (within 2 $\Delta$AICc from the lowest AIC$_c$ model) and global models, with model evaluation parameters, used to assess the effects of ecological and anthropogenic local conditions on the magnitude of lion responses (SMD and RR) to human disturbance.**

| Mixed-effects models | AIC$_c$ | $\Delta$AIC$_c$ | Model weight | $\hat{\tau}^2$ 95% CI | $I^2$ 95% CI | Average ES (95% CI) |
|---|---|---|---|---|---|---|
| **Spatial responses** | | | | | | SMD$_w$ |
| $HFI_{sp} + NDVI_{sp} + NDVI_{tm}$ | 52.95 | 0 | 0.45 | 0.108–0.449 | 82.2–95.0% | −0.268 (−0.433, −0.103)* |
| $HFI_{sp} + CAT_a + NDVI_{sp} + NDVI_{tm}$ | 53.34 | 0.38 | 0.37 | 0.099–0.432 | 80.6–94.8% | −0.279 (−0.437, −0.121)* |
| $HFI_{sp} + NDVI_{sp}$ | 54.79 | 1.83 | 0.18 | 0.132–0.491 | 85.0–95.5% | −0.254 (−0.437, −0.071)* |
| Global model | 59.58 | 6.63 | | 0.105–0.474 | 80.3–94.9% | −0.267 (−0.431, −0.103)* |
| **Temporal responses** | | | | | | RR$_w$ |
| $HFI_{sp} + CAT_a + NDVI_a + NDVI_{sp}$ | −6.29 | 0 | 0.32 | 0.007–0.251 | 66.7–98.7% | −0.068 (−0.104, −0.033)* |
| $HFI_{sp} + CAT_a + NDVI_{sp}$ | −6.12 | 0.17 | 0.29 | 0.005–0.219 | 73.1–99.1% | −0.075 (−0.115, −0.036)* |
| $CAT_a$ | −5.42 | 0.87 | 0.20 | 0.006–0.183 | 81.7–99.3% | −0.089 (−0.142, −0.036)* |
| $CAT_a + NDVI_{sp}$ | −5.27 | 1.02 | 0.19 | 0.005–0.189 | 76.4–99.2% | −0.070 (−0.119, −0.021)* |
| Global model | 5.84 | 12.13 | | 0.006–0.293 | 61.2–98.7% | −0.026 (−0.066, 0.013) |

SMD$_w$ = average weighted standardized mean difference (REM intercept), RR$_w$ = average weighted log response ratio (REM intercept); $HFI_{sp}$ = spatial variation in human footprint index, $CAT_a$ = average cattle production, $NDVI_{sp}$ = average spatial variation in NDVI, $NDVI_{tm}$ = temporal variation in NDVI, $NDVI_a$ = average overall NDVI.
*Coefficient 95% CI significantly different from 0.

also more likely to be detected via direct observation of lions ($F = 9.49$, $df = 2$, $p = 0.002$; Fig. S4). However, this pattern was largely driven by a single study that observed the largest difference in lion nocturnality between low vs. high disturbance areas which was one of few studies that relied on direct observation as opposed to GPS collars or camera traps. The exclusion of this outlier (Dixon test, $Q = 0.454$, $p = 0.02$) eliminated the statistical significance of these effects (median date: $F = 0.145$, $df = 15$, $p = 0.709$; observation type: $F = 2.59$, $df = 2$, $p = 0.11$), and so this result is considered unreliable and disregarded for the remainder of the study.

## Discussion

At over two-thirds of the study sites, lions constrained their spatiotemporal niche to avoid humans by reducing their use of high human disturbance areas or limiting their daytime and crepuscular activity (Fig. 2a). The prevalence of these risk-averse behaviors highlights the tremendous scale of human effects on lions across a wide range of anthropogenic and ecological conditions, including in intensively managed reserves. Our results align with known patterns of human impacts on the spatio-temporal activity of mammals[10,25], including the avoidance of human-dominated areas by large felid species such as the Eurasian lynx[43]. However, synthesizing lion responses across their range revealed that avoidance of humans is not uniform across lion populations. At almost one-quarter of the study sites, lions selected high-disturbance areas more often than low-disturbance areas. Our results indicate that lions are less likely to avoid human disturbance at sites with limited resource availability or fragmented habitats, possibly because resource limitation necessitates the expansion of their spatiotemporal niche (Fig. 3d, e). We also found that lions displayed stronger avoidance responses at sites where intensive cattle production could increase the risks of human-lion conflict (Fig. 3c). Overall, expanding human impacts and environmental changes across the range of lions threaten to downgrade their trophic impacts and are likely to intensify conflict with humans that is a primary threat to lion persistence, though we are unable to empirically connect these behavioral changes to their possible consequences for population dynamics[51,52,61].

Lions at sites with lower average NDVI or higher seasonal variation in NDVI – signaling limited or inconsistent primary productivity – were less likely to avoid human disturbance (Fig. 3d, e). Similarly, we found that increased variation in the human footprint, which could indicate fragmented suitable habitats and resources, contributed to diminished human-avoidance behaviors (Fig. 3a). Highly productive ecosystems that provide high-quality habitat and could support abundant wild prey populations, in contrast, led to spatiotemporal avoidance of human settlements by lions[59,60]. Where resources are scarce or heterogeneously distributed, lions have been shown to expand their home ranges to meet their resource needs[62,63]. Our results indicate that niche expansion in response to spatial and temporal resource scarcity is consistent across lion populations and suggest the risks incurred by lions when encroaching on human-dominated areas can be outweighed by their metabolic needs. Additionally, the responses of lions to resource redistribution and degradation due to climate change and expanding anthropogenic land uses will likely expand the human-lion interface and amplify conflict, a result that highlights the synergy among various threats facing lion populations across their range[64,65]. Climate changes across Africa are projected to exacerbate resource stress for humans and wildlife alike[48,52,66,67], yet the effects of climate change are not usually emphasized in lion and other large carnivore conservation and threat assessments[49,68,69] (but see Carter et al.[70]). Our findings expand upon previous calls to dedicate adequate funding and management capacity to protected areas harboring lions[49,71], while engaging with and empowering local communities to invest in conservation initiatives[72,73]. In particular, we emphasize the need to protect areas that are projected to be refugia from both climate risks and human expansion, as well as corridors that connect suitable lion habitats[46,74].

Highly varied NDVI across a landscape, which could signal increased habitat structure that can diversify the available niches for herbivore prey species (e.g., habitat for both browsers and grazers), produced stronger human-avoidance behavior on both the spatial and temporal axes (Fig. 3b). Habitat structure can also provide optimal hunting grounds for lions[75,76]. Thus, spatial variation in primary resources could improve habitat quality for lions and their prey. Management that encourages diverse habitat structures in protected areas, such as preventing woody encroachment in savannas to maintain intermediate levels of woody cover, could benefit lion populations by supporting more diverse and abundant assemblages of wild prey[77]. Heterogeneous NDVI patterns could alternatively be interpreted as less reliable

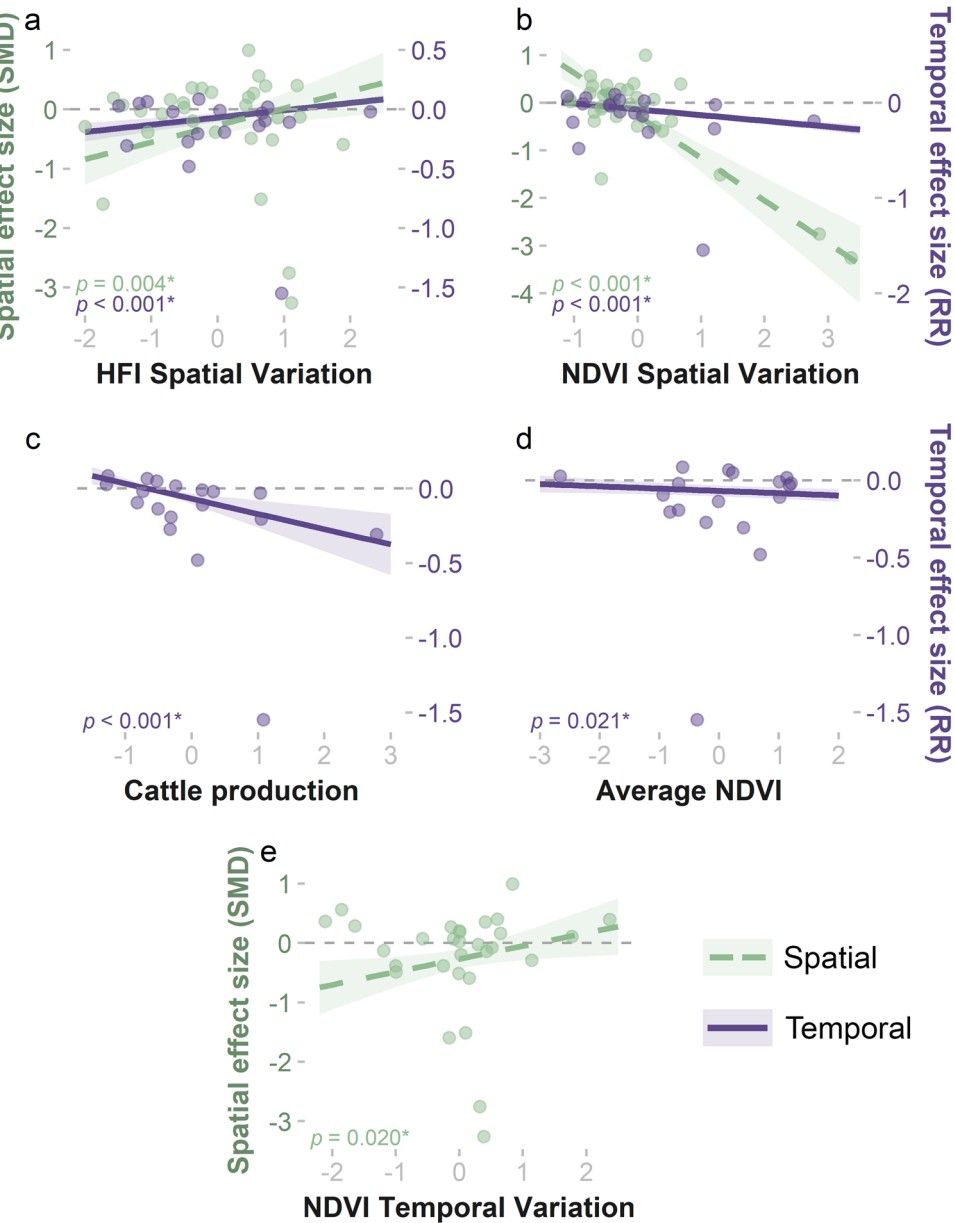

**Fig. 3 The effects of site-level ecological conditions and anthropogenic disturbance on lion spatial (green dashed line, left y-axis) and temporal (purple solid line, right y-axis) responses.** Plots (**a**)–(**e**) depict results from the lowest $AIC_c$ mixed-effects models for both types of responses: **a** spatial variation in the Human Footprint Index ($HFI_{sp}$); **b** spatial variation in NDVI ($NDVI_{sp}$); **c** cattle production ($CAT_a$); **d** average overall NDVI ($NDVI_a$); and **e** temporal variation in NDVI ($NDVI_{tm}$). Linear fits represent the estimated model coefficients with 95% CIs (see Table S3 for variable coefficients). All variables are scaled and centered, and all significantly affected the magnitude of lion responses to human disturbance ($p < 0.05$). Negative values on both y-axes suggest lion avoidance of human disturbance in time and space. SMD = standardized mean difference; RR = log response ratio.

forage resources to support prey species[56]. However, our other model results indicate that less reliable resource availability leads to more risk-taking behaviors by lions, and thus the observed increase in human avoidance given more heterogeneous NDVI does not support this interpretation (Fig. 3d, e).

At sites with higher cattle production, we found that lions increased their nocturnal behavior in response to human disturbance (Fig. 3c). The shift of lions to more nocturnal activity might signal that lions are targeting livestock as prey at these sites because lions are specialized for nighttime hunting and doing so diminishes the risks of encountering humans. Livestock depredation is a primary driver of conflict between lions and humans, which can result in retaliatory killing of lions and threatens lion population growth as well as food and economic security for

human communities[13,65,69]. Alternatively, avoiding humans at these sites may suggest that high-intensity cattle production is exacerbating the disturbances caused by human activities or could reflect prevalent commercial farming that is more likely to exclude predators with fencing and other infrastructure. Though lions prefer hunting at night, they commonly use crepuscular and daytime periods for hunting, feeding, and traversing the landscape, and so lions may be forfeiting access to resources by constricting their temporal niche in response to humans[18,21,78]. Livestock production can degrade habitats for the wild ungulates that lions prefer as prey items[33,79] by monopolizing grazing resources and waterholes as well as amplifying the spread of diseases[70,80]. Where humans accompany grazing livestock herds across large swaths of the landscape during the day, the risks of

human encounters and displacement of wild prey from grazing areas may cause lions to avoid both humans and livestock activity during daytime hours. Spatiotemporal avoidance of grazing livestock has been observed in other large carnivore species, such as snow leopards (*Panthera uncia*)[81] and tigers (*Panthera tigris*)[82], demonstrating that risk-avoidance behaviors are common, though not ubiquitous[82]. Whether lions perceive livestock to be a prey resource or a habitat disturbance, collective action within pastoralist communities is likely among the most important steps toward ameliorating negative lion-human interactions. Residents can simultaneously protect their livelihoods and improve lion habitats and survival through proactive husbandry practices[83] and community-based conservation efforts that create co-benefits for people and wildlife[84,85].

Notably, we did not find evidence to suggest that lions in fenced reserves are less responsive to disturbance (Fig. S4). Many intensively managed fenced reserves, particularly in South Africa, house relatively high-density lion populations that could be shaped by intraspecific interactions to a greater extent than free-roaming populations[71]. Fences create a distinct separation of wildlife from human impacts and should reduce the direct risks posed by humans, which could reasonably lead to higher use of areas near reserve edges and a broader spatiotemporal niche. However, our analyses indicate similar responses to human disturbance by lions in fenced and unfenced reserves, suggesting that the reduction of risk from human disturbance does not necessarily lead to diminished avoidance of humans by lions. Our results may indicate that environmental conditions that govern resource availability in space and time are more important in driving lion spatiotemporal behaviors than the risks posed by human disturbance itself.

Our results demonstrate that human disturbances constrain the realized spatiotemporal niche of lions throughout much of their range. Expanding habitat degradation due to human activities and climatic variability also threatens the capacity of lions to avoid humans and the risks associated with them. As wildlife monitoring efforts continue to expand, future work can build on these findings by 1) explicitly incorporating local-scale interactions that were beyond the scope of this analysis, and 2) empirically connecting the behavioral responses of large carnivores to their consequences for population dynamics. For example, inter- and intraspecific effects within large carnivore communities could cause lion population declines when disturbance favors generalist competitors[86], or lead to more risk-taking behaviors when subordinate individuals are pushed into lower-quality, high-disturbance habitats. Declining resources that limit carrying capacities or maladaptive risk-taking behaviors that provoke human-wildlife conflict could also reduce population sizes to unsustainable numbers. Site-specific estimates of wild prey availability could similarly improve our ability to explain lion responses to human disturbance, as the link between primary productivity and wild prey populations can be decoupled by management strategies or poaching pressures[87,88]. There may also be interactions among anthropogenic and environmental conditions at the site level, such as the transition of livestock from a disturbance that reduces habitat quality to an attractive resource for lions when primary productivity (and by extension, the availability of reources) declines.

Expanding human impacts will reduce suitable habitat and resource availability for wildlife worldwide, and our results indicate that predators will increasingly access more disturbed areas to acquire adequate resources, possibly including targeting livestock as prey items. Human-carnivore conflict is already a primary cause of predator declines worldwide, and our results suggest that the interface between predators and humans will increase in coming years and potentially exacerbate large carnivore population declines[89,90]. Large carnivores and other wildlife usually seek to avoid overlap with humans in space and time[10,25,43], but their ability to do so likely depends on access to relatively stable and predictable environments. Where avoidance of humans is infeasible, carnivores likely face increased risks from human-caused mortality and inadequate habitat quality to support viable populations[17]. In the face of human-driven global change that will intensify environmental variability, successful large carnivore conservation and sustainable coexistence with humans could hinge on the protection and connection of resource-rich habitats and refugia from human disturbances.

## Methods and materials

**Literature search**. We conducted a systematic literature search for peer-reviewed published studies using the ISI Web of Science (WoS) database on lion spatiotemporal activity patterns in relation to human disturbances[91,92]. We also searched for unpublished studies and datasets using the ProQuest Dissertations & Theses database and the Dryad Digital Repository. On WoS and ProQuest, the following Boolean search strings were used: ("Panthera leo" OR "African lion" OR "Lion") AND ("human" OR "anthropogenic") AND ("Avoid*" OR "Space use" OR "Spatial" OR "Respon*" OR "Behavior" OR "temporal" OR "diel" OR "Land use" OR "Management"). We filtered our search for articles (WoS) and dissertations/theses (ProQuest) that fell into the relevant subject categories (e.g., ecology, environmental science, biodiversity conservation) and were published between 1990 and 2021. To expedite the review process, we also excluded results that included terms such as "mountain lion," "pinniped," "tamarin," or "primate" in the abstract or title, as these were common topics in studies returned from the search that were not relevant to our meta-analysis (exact search strings for each database can be found in appendix A). We broadened our dataset by searching the titles of literature cited in each of the included publications (i.e., a snowball approach). Because some studies collected lion activity data but did not report results relevant to this meta-analysis or publish their raw data, we also contacted authors and requested unpublished data to reduce publication bias.

**Inclusion criteria**. We screened the search results first by title, then by abstract, and finally by reading the full text and supplemental materials (Fig. S1). We began by including any study with a title that was related to large carnivores, African wildlife, and/or anthropogenic pressures. We then screened the abstracts of the remaining publications, including those that appeared to measure the activity of large African mammals – or lions specifically – and that might reasonably include considerations of human disturbance in the study. In our final dataset, we included any study that measured lions' spatial or temporal activity across spaces or times of varying human disturbance, enabling a calculation of means and standard deviations of lion activity levels between dichotomous control (low disturbance) and treatment (high disturbance) designations.

Lion spatiotemporal activity was measured by camera traps, VHF or GPS telemetry, or direct observation. We considered studies that measured human disturbance with metrics representing direct human activity or infrastructure (e.g., human detections on camera traps, distance to villages). We also included studies that did not measure human disturbance but provided georeferenced data on lion activity that could be compared to available human disturbance data (Human Footprint Index[54]). When a given dataset of lion spatiotemporal

activity was published in multiple studies, we selected the study that provided the most comprehensive and recent version of the dataset.

**Extraction of lion spatiotemporal data**. For each study or dataset, we calculated the average measure of lion space use (e.g., occupancy estimates, camera trap success, density of observations, or GPS fixes) and/or the proportion of active (i.e., not resting) nocturnal observations ($P$) between designations of low and high human disturbance within the study site (Fig. S2). We extracted measures of lion activity directly from the text, figures, or tables of a published study's results when available, and otherwise used raw data provided with the publication or obtained from study authors to manually calculate measures of lion activity (Fig. S2). (Additional study information and descriptions of study-specific data extraction are available in Tables S1 and S2).

In cases where direct observation surveys of lions occurred primarily during the daytime ($n = 3$), we excluded these studies from calculations of the proportion of nocturnal activity and spatial responses may be biased towards daytime lion activity. We defined nocturnal activity as observations of lion activity that occurred when the sun was lower than six degrees below the horizon unless otherwise specified by the study text[93]. Solar positions were calculated for the time of each lion observation and the specific sample unit coordinates, when available, or the survey site centroid. If the exact site coordinates were not provided by the authors, we estimated the approximate latitude and longitude of the site centroid using site descriptions, figures and maps from the original publications, and Google Maps. To identify 'active' observations in GPS telemetry datasets, we used the sequential clustering algorithm in the "GPSeqClus" R package[94] to identify 'clusters' of more than 2 lion GPS locations given a 100-m search radius and temporal window of 2 days. We considered 'clusters' of GPS fixes to represent lion inactivity that were excluded from our analyses, designating the remaining GPS fixes as active observations that were used to calculate nocturnality. Similarly, we considered all independent camera trap detections to represent active observations. Additional descriptions of data extraction methods can be found in Table S2.

**Quantifying human disturbance**. Designations of human disturbance at the lion observation level (used to calculate effect sizes) were derived in one of two ways depending on the study site and survey design. 1) We used disturbance metrics provided by the study data which we classified as representing infrastructure (such as the distance of observations from villages) or direct human activity (such as humans captured on camera traps). 2) For studies that did not provide disturbance data, we extracted the Human Footprint Index[54] at georeferenced lion observation locations. The human footprint index (HFI) is a global dataset (~10-km resolution) that aggregates various axes of human impacts on ecosystems (e.g., population size, infrastructure, agriculture, etc.) into a single index of human pressure, and we thus categorized it as representing both infrastructure and direct human activity[54]. Both approaches were used to identify locations or times that experienced low versus high levels of disturbance that could be assigned to measurements of lion activity. Low and high disturbance designations could represent discrete periods of time or land units (e.g., land use types or management blocks), or a continuous mosaic of human disturbance (e.g., distance to a village) that could be binned into low and high disturbance categories. In studies with a continuous mosaic of human disturbance, we extracted the value of the

human disturbance metric for each spatial sampling unit (e.g., camera station or grid cell). For datasets consisting of lion GPS fixes without clear spatial sample units, we used the average daily displacement distance ($\bar{\delta}km$) of lions to create a $\bar{\delta}km^2$ grid across the study site to serve as the sample units within which observations were aggregated for subsequent analyses. If $\bar{\delta}km^2$ could not be calculated, such as for direct observation studies, we used a 1-km$^2$ grid. We used a power analysis of a two-tailed t-test to guide our selection of cutoff values to assign low and high disturbance, which indicated that approximately 60 samples or more per group were required to achieve 80% power to detect differences in means. For each study site, we assigned the 1st and 3rd quartiles of the human disturbance variable as cutoff values for low and high disturbance categories, eliminating noise generated by intermediate levels of disturbance. However, if the number of sample units in each quartile was <60 for a given site, we instead used the median value of disturbance at that site as the cutoff value (Table S2).

**Effect sizes**. To evaluate the effects of human disturbance on lion activity, we calculated the standardized mean difference (SMD) and log response ratio (RR) of lion space use and nocturnality, respectively, between low and high human disturbance treatments at each study site using the "*metafor*" R package[91,95]. In determining spatial responses, we calculated SMD using Hedge's $d$ metric of effect size[91,96]. When SMD < 0, human disturbance negatively impacted lion space use, indicating human avoidance behaviors, while SMD > 0 conversely indicated more use of high disturbance areas. We similarly calculated the log response ratio of temporal responses using:

$$RR = \ln\left(\frac{P_{low}}{P_{high}}\right) \tag{1}$$

in which $P_{low}$ and $P_{high}$ represent the proportion of active lion observations that occurred during nocturnal hours in low and high human disturbance areas. Higher levels of lion nocturnality in response to human disturbance are signified by RR < 0, while RR > 0 indicates more diurnal activity in high-disturbance areas. Increased nocturnal activity in high disturbance treatments is assumed to be an avoidance response in large carnivores, as human activity is usually concentrated during daylight hours[25,26,97]. We also report the back-transformed RR to calculate the percent by which lions increased their nocturnality due to human disturbance. We calculated the variance of RR for each study as follows[91], where $n$ indicates sample sizes of lion observations in low and high human disturbance areas:

$$Variance_{RR} = \frac{(1 - P_{low})}{n_{low}P_{low}} + \frac{(1 - P_{high})}{n_{high}P_{high}} \tag{2}$$

**Spatial variables**. We considered three site-level spatial variables across all of the study sites that might influence the strength of lion responses to within-site human disturbance (i.e., the magnitude of meta-analysis effect sizes): 1) cattle production, 2) human footprint, and 3) primary productivity. We created a circular buffer around the geographic centroid of each study site (hereafter, the buffered study area) within which we extracted and summarized the 3 site-level spatial variables. To assess the sensitivity of our analyses to buffer size selection, we compared two buffer methods: one with the buffer area equal to the study area size specified by the study authors (study-specific buffer area), and another applying the minimum study area size to the buffer of all sites (uniform buffer area). The two methods were

compared using univariate model selection, as described in the 'Meta-regression and analyses' section.

We calculated the average cattle production ($CAT_a$) within each circular buffered study area using a dataset that estimates global cattle production (~ 10-km resolution)[98]. We then used the human footprint index to assess the overall human pressure at each site[54]. Because this dataset provides HFI estimates in 2000 and 2019, we extracted the site-level HFI data for the year closest to the median date of each lion survey dataset. We expected that the overall level of human pressure as well as the existence of spatial refugia from those pressures might influence lion habituation, and thus their avoidance behaviors, so we calculated the average ($HFI_a$) and the coefficient of variation (CV) of HFI ($HFI_{sp}$, representing spatial variation in HFI) within each buffered study area. Finally, we used the Normalized Difference Vegetation Index (NDVI) time series dataset provided by MODIS Land Products (250-m resolution) to summarize primary productivity for each study, thereby accounting for environmental changes in resources such as surface water and habitat cover that shape lion foraging strategies as well as the forage quality for herbivore prey[57–60]. Though the correlation between primary production and wild prey availability may be decoupled in protected areas experiencing large-scale defaunation in recent decades[87], we chose to include NDVI in our study as the closest proxy available for broad-scale patterns in site-level resource availability, including wild prey. We obtained NDVI layers for the 1st day of each month from January 2000 (the earliest available date) to September 2019 (the latest date of a lion survey) using the '*MODIStsp*' R package[99]. We calculated 3 metrics of NDVI at each site, where $\mu_i$ and $\sigma_i$ are the average and standard deviations of all NDVI pixels within the buffered study area for month $i$:

1) the overall mean monthly NDVI across months,

$$NDVI_a = \frac{1}{n}\sum_{i=1}^{n} \mu_i \quad (3)$$

2) the average within-month CV of NDVI (representing mean spatial variation in productivity),

$$NDVI_{sp} = \frac{1}{n}\sum_{i=1}^{n} \frac{\sigma_i}{\mu_i} \quad (4)$$

and 3) the CV of mean monthly NDVI (representing temporal variation in mean monthly productivity),

$$NDVI_{tm} = \frac{\sqrt{\frac{\sum(\mu_i - NDVI_a)^2}{n}}}{NDVI_a} \quad (5)$$

Because lion responses to humans could be influenced by long- and short-term patterns in site-level ecosystem productivity that influence wild prey abundances, we compared the sensitivity of our analyses to the temporal scale of NDVI layers used in calculating these metrics. Thus, the 3 metrics were calculated at each study site once using the entire January 2000-September 2019 monthly NDVI dataset (long-term) and again using only the months in which each lion study took place (short-term). The two temporal ranges for NDVI metrics were compared using univariate model selection, as described in the "Meta-regression and analysis" section. All spatial variables were scaled and centered to produce standardized model coefficients in the statistical analyses, allowing for comparisons among variable effects on lion responses to human disturbance. We also assessed the correlation among all of the spatial variables to ensure that highly correlated variables ($r > 0.6$) were not included together in statistical models. To support our inference of NDVI as a metric for climatic trends, we also extracted monthly rainfall estimates for each study site (excluding 2 sites in India due to data availability) from

TAMSAT precipitation data[100] for Africa (2000–2019) and created a linear model which compared the effects of average precipitation on average NDVI per month for each study site (i.e., a site:rainfall interaction term). Because the two variables are highly correlated (Table S4, Fig. S2) and NDVI more directly influences wild prey availability for lions[59,60], we did not include precipitation data in subsequent analyses.

**Statistics and reproducibility: Meta-regression and analyses.** For studies that measured both response types, we modeled spatial effect sizes as a function of temporal effect sizes using a linear model to assess the existence of spatiotemporal tradeoffs in human avoidance behaviors. We then assessed how the heterogeneity in lion responses to human disturbance among studies (i.e., the effect sizes) was influenced by the metrics calculated from the three spatial variables using separate mixed-effects models for spatial and temporal lion responses (with the study ID as a random effect). We weighted the studies using the inverse of their calculated sampling variance and used the maximum likelihood estimation of residual heterogeneity. The intercept term of the generated models was interpreted as the average weighted effect size of human disturbance on lion responses ($SMD_w$ or $RR_w$). We used the Akaike Information Criterion corrected for small sample sizes ($AIC_c$) to compare models and identified the model with the lowest $AIC_c$ as the best performing model during model selection. All models were built and assessed using the '*metafor*' package in R[95].

We first compared the performance of each spatial variable calculated using the two buffer sizes (all spatial variables) and two temporal ranges (for NDVI) by creating and comparing univariate mixed-effects models. The metric included in the lowest $AIC_c$ model for each spatial variable metric (e.g., $HFI_{sp}$) was then chosen to be included in the global model. If the models did not differ by > 2 $\Delta AIC_c$, then we used the metrics calculated based on the site-specific study area size and study period. Our results were robust to the selection of buffer size or temporal range of NDVI data except for one case in which the minimum buffer size offered marginally higher explanatory power for temporal responses to spatial variation in primary productivity ($NDVI_{sp}$, $\Delta AIC_c = 2.19$). We then compared mixed-effects models using all combinations of the 6 spatial variable metrics included in the global model ($SMD$ or $RR \sim CAT_a + HFI_a + HFI_{sp} + NDVI_a + NDVI_{sp} + NDVI_{tm}$). We present models with $\Delta AIC_c < 2$ compared to the best. We used the standardized variable coefficients from the lowest $AIC_c$ model to assess variable effects. Mixed-effects models with small sample sizes can result in inflated and unreliable values of common model evaluation metrics[101,102], such as $\hat{\tau}^2$, $I^2$, and $R^2$, which estimate the amount of heterogeneity in the true effect sizes, the proportion of variability attributed to heterogeneity among the true effect sizes, and the amount of heterogeneity explained by model variables, respectively. We thus evaluated the performance of the mixed-effects models at explaining heterogeneity in lion responses using the 95% confidence intervals of the $I^2$ and $\hat{\tau}^2$ measures, rather than the precise point estimates, as well as the $Q_E$-test for residual heterogeneity. For evaluation of publication bias in the top-performing models, see Supplementary Information 1 and Fig. S6.

Finally, we assessed whether study characteristics — including the type of human disturbance measured to calculate effect sizes (e.g., infrastructure versus direct activity), whether the study site is fenced, lion observation method, study area size, study season, study duration, and median study date — might influence the observed effect sizes using ANOVA tests for categorical variables and linear models for continuous variables.

**Reporting summary**. Further information on research design is available in the Nature Portfolio Reporting Summary linked to this article.

## Data availability

All effect sizes and study information is available in the Supplementary Tables. Data tables for meta-regression and statistical analyses, including effect sizes and extracted spatial variables, are provided as Supplementary Data. All other data or information are available from the corresponding author upon reasonable request.

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

## Acknowledgements

The lion monitoring data used in this study are the product of massive fieldwork and organizational efforts from numerous research groups, field teams, and study systems across the contemporary range of lions. We thank all of the research teams and contributors of each of the studies used here for their efforts and for transparent data reporting that enabled their inclusion in our meta-analysis. We also thank I. Sankaran for help with the preliminary literature review. Finally, we are grateful for the editors and anonymous reviewers who provided constructive feedback and helpful comments on this manuscript.

## Author contributions

Conceptualization: K.L.M., N.J.S., N.H.C. Data Analysis: K.L.M., with input from N.J.S. and N.H.C. Data Contribution: J.L.B., M.B., E.D., K.T.E., R.F., D.S.G., M.W.H., K.E.H., F.G.T.R., G.S., J.P.S., L.K.W., C.C.W. Writing—original draft: K.L.M. Writing—review & editing: All authors.

## Competing interests

The authors declare no competing interests.
