## [Peer Review File · Communications Biology]

Reviewers' comments:

Reviewer #1 (Remarks to the Author):

The article titled as 'Tradeoffs between resources and risks shape large carnivore responses to human disturbance' is a very interesting and well done work describing how human disturbance affects lion behavior. It is a meta-analysis performed with data from 23 studies, and results indicate that lions reduce the use of space and increase nocturnal activity where human disturbance is high. Moreover, this trend seems to be balanced with the need for resources. Overall, I think the manuscript is well developed and written, and results clear and convincing. I only have a few comments of minor importance. 1) I think the title is not very informative, given that authors actually focus on lions, not on large carnivores (i.e. also considering other species). So, I suggest to modify the title to focus on the studied species; 2) In relation to cattle production, lions show an increased nocturnal behavior, and then they avoid humans. But in doing so, they also avoid cattle, right? Is it not an adaptation to avoid both (humans and cattle) and thus to reduce potential conflicts? Perhaps authors could estimate number of attacks to livestock as proxy of conflicts and include this variable into the analyses; 3) Indeed, and as authors recognize, the lion is a nocturnal animal in a great extent. In fact, likewise other felid species, they prefer to hunt at night. So, I feel this aspect of the lion's biology has been not completely dealt in the Discussion section; 4) I miss something directly related to conservation: what are the effects of reduced space use and increased nocturnal activity on lion populations? Are they declining just where those effects have been found? This is important because, particularly for nocturnal behavior, it could be possible that the described effects have no consequences on population dynamics; and 5) I find repetitive the last paragraph of Introduction (lines 105-126), which should belong to the Mat-Met section. However, I understand this may be a weird result of leaving the Mat-Met section at the end of the manuscript. Indeed, if Mat-Met section were the second one, such as normal/usual for the structure of a scientific article, the indicated paragraph would not be necessary. I definitively disagree with this altered structure of scientific papers, as well as with numeric citation system, which makes more difficult the assessment of used literature. Nevertheless, I understand that authors have nothing to do with this last comment. Congratulations for this nice work.

Reviewer #2 (Remarks to the Author):

Review Report

Manuscript: Tradeoffs between resources and risks shape large carnivore responses to human disturbance

Comments: I commend the authors on an impressively neat piece of study. I thoroughly enjoyed reading the manuscript, and such large-scale meta-analysis is imperative and powerful to comprehend species ecology in the Anthropocene. Well done! However, I do have a few comments which I would kindly implore the authors to consider and perhaps incorporate

1. While human-activity does impede carnivore space and time use globally, many carnivore species use this 'disturbance' to their benefits. Human-subsidized food in terms of (relatively) easily catchable prey and carcass dumps, making dens near human settlements to reduce competition for stronger predators (case in point dholes and tigers in C. India), to name a few. I think adding this angle to the intro or discussing this might add value to the depth of the manuscript and would set up the tone for the analysis.
2. Gogoi et al.'s Gir paper (the only inclusion of Asian lions) is based inside the park that has limited human-lion interactions. Outside the park, lions and humans coexist at varying densities, and lion activity and spatial extent are modulated by the duration of human-lion exposure. Human communities that have been exposed to lions for generations show more tolerance and lion activity patterns and human-avoidance are far lower than in areas where they have colonized recently. I wonder if this might be the case across many other African sites.

3. While the authors have estimated the spatial and temporal trends modulated by a few predictor variables, I think adding some other crucial human social predictors such as socio-economic background, cultural tolerance would help understand the finer nuances of lion-human coexistence. Currently the trends seem very mechanistic and based on only a few predictors
4. Why were other spatial variables such as DEM and terrain brokenness (seasonal rivers) not considered? In my experience, broken terrain and terrain ruggedness can provide important micro-habitat features that can have strong effect on site and time use by lions in human-dominated landscapes.
5. Is there cultural/trophy hunting of lions in any of the sites that have been included in the study? How would that impact lion temporal activity?
6. The discussion can use inclusion of examples and comparison with other similar studies and species

I again commend the authors on a great paper - inclusion of the above can make the paper even stronger. Best Wishes, Stotra

Reviewer #3 (Remarks to the Author):

Hi Team

Apologies for the lateness of my review. This seems like an important paper and I recommend it for publication pending a few fairly straightforward revisions.

How exactly were these lion data obtained, or more importantly, what are they? Are they raw datasets that you've asked for from study authors on gps locations? Are they summaries from papers? This is not explicitly made clear in the methods but all of your results rest on this...

In another way: "we calculated the average measure of lion space use (e.g., 332 occupancy estimates, camera trap success, density of observations, or GPS fixes)" - what is this? Did you pull the points from the maps in the papers or get raw data from the authors of those papers?

Not a fan of your prey availability data NDVI - some of the earth's most productive systems by NDVI metrics are the most heavily poached...the more I read this now the more I don't like this component of the study. NDVI is not going to be a great proxy for prey availability - is there not better prey data you can get from your respective sites? This is a major shortcoming of this study and needs to be addressed.

I guess though do you actually need prey availability? Are you not interested in lion space use in relation to human disturbance...can you find data on prey availability or a better proxy thereof? NDVI to me doesn't make sense!

The more I read this NDVI stuff the more it sounds like the Martin and de Meleunar leopard density prediction paper - really poor science. Wildlife can be removed without doing anything to the habitat therefore NDVI is BAD! You should read Alan Rabinowit'z How to Drive a Species to Extinction: the case of the Sumatran Rhino - in there you will find exactly my point!

Overall though I still like this paper a lot as it clearly shows lions are pressured by human disturbance.

I'm asking for another proxy, another data source for prey availability - something!

That's really my only concern

Good luck with this fairly small revision.

Alex Brackowski

Reviewers' comments:

Reviewer #1 (Remarks to the Author):

The article titled as 'Tradeoffs between resources and risks shape large carnivore responses to human disturbance' is a very interesting and well done work describing how human disturbance affects lion behavior. It is a meta-analysis performed with data from 23 studies, and results indicate that lions reduce the use of space and increase nocturnal activity where human disturbance is high. Moreover, this trend seems to be balanced with the need for resources. Overall, I think the manuscript is well developed and written, and results clear and convincing. I only have a few comments of minor importance.

Mills et al.: Thank you for your thoughtful feedback, we are very glad to see such a positive review.

1) I think the title is not very informative, given that authors actually focus on lions, not on large carnivores (i.e. also considering other species). So, I suggest to modify the title to focus on the studied species.

Mills et al.: We have changed the title from "...shape large carnivore responses..." to "...shape the responses of a large carnivore..." to reflect that it is a species-specific study. We have opted not to specify the species name in the title because we want to indicate that the observed patterns likely apply to the broader taxon, rather than overly narrow the scope of the work to the reader. The ways in which the scope and results of this study relate to other large carnivore species are discussed throughout the Introduction and Discussion sections.

2) In relation to cattle production, lions show an increased nocturnal behavior, and then they avoid humans. But in doing so, they also avoid cattle, right? Is it not an adaptation to avoid both (humans and cattle) and thus to reduce potential conflicts? Perhaps authors could estimate number of attacks to livestock as proxy of conflicts and include this variable into the analyses.

Mills et al.: Records of livestock attacks are unfortunately not available or accessible at all of the study sites in the meta-analysis, so we are unable to include this additional piece in our analyses. We have revised the sentence in line 256 of the discussion to make the idea that lions may be avoiding cattle more explicit.

Lines 254-257: Revised text (underlined text are added revisions): "*Where humans accompany grazing livestock herds across large swaths of the landscape during the day, the risks of human encounters and displacement of wild prey from*

grazing areas may cause lions to avoid both humans and livestock activity during daytime hours."

3) Indeed, and as authors recognize, the lion is a nocturnal animal in a great extent. In fact, likewise other felid species, they prefer to hunt at night. So, I feel this aspect of the lion's biology has been not completely dealt in the Discussion section.

Mills et al: This topic is discussed in lines 248-251 in the Discussion, but we have revised to make it more thorough. We included an additional reference (Suraci et al. 2019, *Ecology*) to support the idea that lions use non-nocturnal times of day for important activities like feeding and traveling, despite a preference for nighttime hunting.

Lines 248-251, revised text: "Though lions prefer hunting at night, they also use crepuscular and daytime periods for hunting, feeding, and traversing the landscape. As a result, lions may be forfeiting access to resources by constricting their temporal niche in response to humans^{18,21,81}."

Original text: "Though lions are primarily nocturnal, they commonly exploit hunting opportunities during crepuscular and daytime periods and may be forfeiting access to important resources by constricting their temporal niche in response to humans^{18,21}."

4) I miss something directly related to conservation: what are the effects of reduced space use and increased nocturnal activity on lion populations? Are they declining just where those effects have been found? This is important because, particularly for nocturnal behavior, it could be possible that the described effects have no consequences on population dynamics.

Mills et al.: This is another case in which the data are not available at a large enough scale to evaluate how these behavioral changes are connected to population dynamics. The only reliable and comparable indicator of lion population status that we are aware of is the population growth parameter estimates published by Bauer et al. 2015, *PNAS*, but these were not calculated for almost 40% of our study sites and generally exhibit little variation which would limit our inference (range of 0.96-1.3). We did consider including this analysis early on, but ultimately chose not to because it would further reduce our sample size.

We have added the following sentence of the Discussion paragraph 1 in lines 202-203: "However, we were unable to connect these behavioral changes to the population dynamics of lions in this study⁶⁴." We are careful throughout not to make the claim that behavioral changes directly influence population viability.

We have also revised/added the following sentences near the end of the Discussion (lines 282-284, 287-289) that highlight the need to make the connection between behavioral responses to humans and their impacts on population dynamics, while reminding the reader that we aren't able to extrapolate population dynamics from our results:

Revised, lines 282-284 (underlined text are added revisions): *"As wildlife monitoring efforts continue to expand, future work can build on these findings by 1) explicitly incorporating local-scale interactions that were beyond the scope of this analysis, and 2) empirically connecting the behavioral responses of large carnivores to their consequences for population dynamics."*

New sentence, lines 287-289: *"Declining resources that limit carrying capacities or maladaptive risk-taking behaviors that provoke human-wildlife conflict could also reduce population sizes to unsustainable numbers."*

5) I find repetitive the last paragraph of Introduction (lines 105-126), which should belong to the Mat-Met section. However, I understand this may be a weird result of leaving the Mat-Met section at the end of the manuscript. Indeed, if Mat-Met section were the second one, such as normal/usual for the structure of a scientific article, the indicated paragraph would not be necessary. I definitively disagree with this altered structure of scientific papers, as well as with numeric citation system, which makes more difficult the assessment of used literature. Nevertheless, I understand that authors have nothing to do with this last comment.

Mills et al.: The last paragraph of the introduction is indeed repetitive of the methods because of the journal's desired structure, and we have made no changes here.

Congratulations for this nice work.

Mills et al.: Thank you again.

Reviewer #2 (Remarks to the Author):

Review Report

Manuscript: Tradeoffs between resources and risks shape large carnivore responses to human disturbance

Comments: I commend the authors on an impressively neat piece of study. I thoroughly enjoyed reading the manuscript, and such large-scale meta-analysis is imperative and

powerful to comprehend species ecology in the Anthropocene. Well done! However, I do have a few comments which I would kindly implore the authors to consider and perhaps incorporate.

Mills et al.: Thank you very much for your encouraging and constructive comments.

1) While human-activity does impede carnivore space and time use globally, many carnivore species use this 'disturbance' to their benefits. Human-subsidized food in terms of (relatively) easily catchable prey and carcass dumps, making dens near human settlements to reduce competition for stronger predators (case in point dholes and tigers in C. India), to name a few. I think adding this angle to the intro or discussing this might add value to the depth of the manuscript and would set up the tone for the analysis.

Mills et al.: We have included more explicit consideration of this angle in the Introduction section by adding the following sentence in lines 72-75: "Conversely, some species exploit resource subsidies provided by humans by accessing human-dominated areas at times of low risk, but this can lead to intensified human-carnivore conflicts over food resources like livestock^{32,33}."

2) Gogoi et al.'s Gir paper (the only inclusion of Asian lions) is based inside the park that has limited human-lion interactions. Outside the park, lions and humans coexist at varying densities, and lion activity and spatial extent are modulated by the duration of human-lion exposure. Human communities that have been exposed to lions for generations show more tolerance and lion activity patterns and human-avoidance are far lower than in areas where they have colonized recently. I wonder if this might be the case across many other African sites.

Mills et al.: It is possible (maybe likely) that this is the case in African sites, but we do not know of any datasets that would enable the comparison of the duration of lion and human co-occurrence across our study sites. These time periods likely extend back hundreds or thousands of years at the very least. We discuss the idea of habituation to human disturbance briefly in the introduction (line 83-84), and our use of HFI as an explanatory variable does capture spatial variation in human pressures (including population density). But as we cannot reliably estimate how long populations have been co-occurring with humans, this addition would be outside the scope of the study.

3) While the authors have estimated the spatial and temporal trends modulated by a few predictor variables, I think adding some other crucial human social predictors such as socio-economic background, cultural tolerance would help understand the finer

nuances of lion-human coexistence. Currently the trends seem very mechanistic and based on only a few predictors

Mills et al.: Though there are other characteristics of the study areas that could influence lion responses to humans, it is outside the scope of our study to include every potential influence. Indeed, given the general lack of data at the scale of our study, it was not feasible to include every possible study area characteristic suggested here. Furthermore, the primary objective of our study was to examine how variables related to global changes might influence lion responses to humans, and so we included only those additional study area characteristics that were necessary to verify that our analyses were not biased by how the individual studies were conducted.

Simply put, we didn't want to over-parameterize the models or reduce their sample sizes, given that we had 23 studies included.

4) Why were other spatial variables such as DEM and terrain brokenness (seasonal rivers) not considered? In my experience, broken terrain and terrain ruggedness can provide important micro-habitat features that can have strong effect on site and time use by lions in human-dominated landscapes.

Mills et al.: As stated above, it is outside the scope of our study to include these additional variables. The study focuses on broad-scale drivers of lion responses related to global change, rather than micro-habitat features that are not directly linked to global change. Though they may provide added explanatory power, they do not relate to the main goals of the paper.

5) Is there cultural/trophy hunting of lions in any of the sites that have been included in the study? How would that impact lion temporal activity?

Mills et al.: Though poaching in some systems may increase the risks of humans at night or in certain areas, data on hunting/poaching intensity is unavailable for most sites in our study. As we cannot reliably compare hunting pressures among sites, any consideration of the impacts of poaching pressure on lion activity is outside the scope of our study.

6) The discussion can use inclusion of examples and comparison with other similar studies and species

Mills et al.: We have included a few additional examples of similar studies/results in the discussion section.

Lines 188-193: "Our results align with known patterns of human impacts on the spatiotemporal activity of mammals^{10,25,61}, including the avoidance of human-dominated areas by large felid species such as the Eurasian lynx (*Lynx lynx*)⁴³ and African leopard (*Panthera pardus*)⁶². With our meta-analytic framework, we also build upon a recent review of African lion habitat selection, which highlighted how anthropogenic landscapes alter expected lion habitat use⁶³."

Lines 257-260: "Spatiotemporal avoidance of grazing livestock has been observed in other large carnivore species, such as snow leopards (*Panthera uncia*)⁸⁵ and tigers (*Panthera tigris*)⁸⁶, demonstrating that risk-avoidance behaviors are common, though not ubiquitous⁸⁶."

I again commend the authors on a great paper - inclusion of the above can make the paper even stronger. Best Wishes, Stotra

Mills et al.: Thank you very much for your kind comments and thoughtful suggestions.

Reviewer #3 (Remarks to the Author):

Hi Team

Apologies for the lateness of my review. This seems like an important paper and I recommend it for publication pending a few fairly straightforward revisions.

Mills et al.: Thank you very much for your comments and positive recommendation.

1) How exactly were these lion data obtained, or more importantly, what are they? Are they raw datasets that you've asked for from study authors on gps locations? Are they summaries from papers? This is not explicitly made clear in the methods but all of your results rest on this... In another way: "we calculated the average measure of lion space use (e.g., 332 occupancy estimates, camera trap success, density of observations, or GPS fixes)" - what is this? Did you pull the points from the maps in the papers or get raw data from the authors of those papers?

Mills et al.: Detailed information on the data pulled for each study is provided in the supplement, but we have added some clarity in this section to point the reader in the right direction if they want more details. The following new sentence clarifies the sources of the data for the included studies, and highlights a supplementary table with study-specific details and a flow chart depicting the data extraction process for additional information:

Lines 352-356: "We extracted measures of lion activity directly from the text, figures, or tables of a published study's results when available, and otherwise used raw data provided with publications or obtained from study authors to manually calculate measures of lion activity (Figure S2). (Additional information on the study-specific data extraction is available in Table S2.)"

2) Not a fan of your prey availability data NDVI - some of the earth's most productive systems by NDVI metrics are the most heavily poached...the more I read this now the more I don't like this component of the study. NDVI is not going to be a great proxy for prey availability - is there not better prey data you can get from your respective sites? This is a major shortcoming of this study and needs to be addressed.

Mills et al.: Unfortunately, the data are unavailable to include direct measures of prey availability for each study site. Even for the studies we include in the meta-analyses, collecting the data on prey availability would be outside the scope of each independent study; the data are challenging to come by, which is why proxies are often used, such as NDVI that is strongly correlated with prey availability in many systems. Most notably, Pettorelli et al. 2009, *Am. Nat.*, found positive relationships between NDVI and African large herbivore densities at the continental scale, including many species that are favored prey of lions. Additionally, Fritz et al. 1994, *Proc. B.*, also found strong positive relationships between primary productivity indicators - particularly rainfall - and large ungulate biomass. In another paper by Mills et al. that is in review at *Ecol. Apps.*, we found a strong positive effect of NDVI on the availability of wildebeest and zebra (see figure below).

Figure from Mills et al., in review at *Ecol. Apps.*, showing significant positive relationship between NDVI measures and wild prey availability (wildebeest and zebra abundance) from logistic regression model.

However, we do recognize and agree that primary productivity is not a perfect measure of prey availability because these can become decoupled in places where defaunation is rampant. We highlight the valuable addition of direct measures of prey availability to future work in lines 287-289: "Site-specific estimates of wild prey availability could similarly improve our ability to explain

lion responses to human disturbance, as the link between primary productivity and wild prey populations can be decoupled by management strategies or poaching pressures^{91,92}.

At the same time, NDVI/primary productivity still conveys information about the ecosystem beyond potential prey availability, particularly the availability of water resources (as we show in the correlation between NDVI and precipitation, Figure S3) and habitat structure that also influence large carnivore activity. We recognize that we may have focused on the link between NDVI and prey availability too strongly in the text. These connections (and their weaknesses) are discussed in the manuscript, but we have made changes throughout to better convey that NDVI is being used as a proxy for resource availability more generally, not only for prey availability. Additionally, though high NDVI may not necessarily cause abundant prey, many of our conclusions are more focused on the opposite case in which low/inconsistent NDVI reduces a system's ability to support abundant wild prey, a relationship that is not as impacted by hunting/poaching pressures.

The most substantial changes to the text are as follows:

Lines 121-128 (underlined text are added revisions): ***"To assess the effects of resource availability on lion avoidance of disturbance, we used satellite-derived measures of vegetation greenness (i.e., Normalized Difference Vegetation Index [NDVI]), including the overall average NDVI during the study period as well as spatial and temporal variability in NDVI. NDVI correlates directly with primary productivity and often predicts wild prey abundances and the availability of other primary resources such as surface water and habitat that strongly influence lion foraging strategies⁵⁶⁻⁶⁰."***

Lines 294-296: ***"...such as the transition of livestock from a disturbance that reduces habitat quality to an attractive resource for lions when primary productivity (and by extension, the availability of resources) declines."***

Lines 441-449 (underlined text are added revisions): ***"Finally, we used the Normalized Difference Vegetation Index (NDVI) ... to summarize primary productivity for each study, thereby accounting for environmental changes in resources such as surface water and habitat cover that shape lion foraging strategies as well as the forage quality for herbivore prey^{56,57,59,60}. Though the correlation between primary production and wild prey availability may be decoupled in protected areas experiencing large-scale defaunation in recent decades⁹¹, we chose to include NDVI in our study as the closest proxy available for broad-scale patterns in site-level resource availability, including wild prey."***

3) I guess though do you actually need prey availability? Are you not interested in lion space use in relation to human disturbance...can you find data on prey availability or a better proxy thereof? NDVI to me doesn't make sense! The more I read this NDVI stuff the more it sounds like the Martin and de Meleunar leopard density prediction paper - really poor science. Wildlife can be removed without doing anything to the habitat therefore NDVI is BAD! You should read Alan Rabinowit'z How to Drive a Species to Extinction: the case of the Sumatran Rhino - in there you will find exactly my point!

Mills et al.: The primary goal of this study was to understand how lions respond to human disturbances in the context of global environmental and anthropogenic changes. Because of the importance of resource availability in shaping lion and other large carnivore home ranges and foraging strategies, we feel that including NDVI as a proxy for resource availability is a fundamental necessity of the paper, especially given the broad spatial scale of our analysis. As noted above, we recognize the need to attenuate the link between NDVI and wild prey availability - since these can be decoupled as exemplified by the Sumatran rhino example - and have revised the text to alleviate this concern but have decided to retain NDVI in the analysis.

Overall though I still like this paper a lot as it clearly shows lions are pressured by human disturbance. I'm asking for another proxy, another data source for prey availability - something! That's really my only concern.

Mills et al.: Thank you for your thoughtful comments, they have certainly helped to strengthen the paper. We are very happy to hear that you liked the paper a lot!

Good luck with this fairly small revision.
Alex Brackowski

REVIEWERS' COMMENTS:

Reviewer #1 (Remarks to the Author):

Thank you very much for your responses, which I find totally convincing. The manuscript has been improved and I think it truly is an important contribution. Congratulations for your work.

Jorge Lozano

Reviewer #2 (Remarks to the Author):

Thank you for revising the manuscript and addressing the questions. I appreciate the detailed response to reviewers' comments :)
I think the manuscript has shaped up nicely and will make a valuable contribution to the field! Well done :)
Cheers, Stotra Chakrabarti

Reviewer #3 (Remarks to the Author):

I'm sorry but I do not agree with the assessments of NDVI and prey availability - this is simply from my extensive fieldwork in Uganda. Your estimates of NDVI will fall apart completely when linked to prey in my field sites where poaching is rife.

I still suggest you remove the prey out of it - bad methods wont solve the prey issues in this paper.

Other than that the new version is fine for publication (again excluding the NDVI!).

Kirby L. Mills, Ph.D.

August 25th, 2023
Communications Biology

Dear Editorial Team,

Thank you for your feedback and invitation to resubmit a final version of our manuscript titled "Tradeoffs between resources and risks shape the responses of a large carnivore to human disturbance". We found the reviewers' previous comments to be very kind and helpful, and we have provided our final responses to their remaining concerns below in bold. We did not include comments from reviewers #1 and #2, as they did not request any additional changes to the manuscript. All line numbers reflect those in the attached document including tracked changes.

In response to the reviewer's remaining concern about NDVI measures for prey availability, we have made some additional minor changes to the text to adequately discuss the limitations of this approach.

We wholeheartedly thank the reviewers for their thoughtful, constructive, and overall positive feedback on this manuscript. We are very excited to publish this work in *Communications Biology*.

Sincerely,

Kirby Mills, Neil Carter, and Nate Sanders

Reviewers' comments:

Reviewer #3 (Remarks to the Author):

I'm sorry but I do not agree with the assessments of NDVI and prey availability - this is simply from my extensive fieldwork in Uganda. Your estimates of NDVI will fall apart completely when linked to prey in my field sites where poaching is rife.

I still suggest you remove the prey out of it - bad methods won't solve the prey issues in this paper.

Other than that the new version is fine for publication (again excluding the NDVI!).

Mills et al.: We appreciate the reviewer's concerns regarding NDVI, and we have further strengthened our language in a few places to reflect that it is not a perfect proxy of prey availability. In addition to our previous revisions, please see revisions in the Introduction (lines 113 and 131-136) and Discussion (lines 215-216). Underlines text indicates new text.

-Line 113, revised text: "3) lion avoidance of human disturbance is mitigated by variable primary productivity that could limit wild prey availability."

-Lines 131-136, original text: "NDVI correlates directly with primary productivity and often predicts wild prey abundances and primary resource availability more generally, such as surface water and habitat that strongly influence lion foraging strategies⁵⁶⁻⁵⁸."

-Lines 131-136, revised text: "NDVI correlates directly with primary productivity and thus provides a measure of primary resource availability, such as surface water and habitat that strongly influence lion foraging strategies⁵⁶⁻⁵⁸. NDVI also generally predicts wild prey abundances across Africa, though this relationship can be decoupled when wild prey populations in productive sites are decimated by overharvesting or other anthropogenic impacts^{59,60}."

-Lines 215-216: "Highly productive ecosystems that provide high-quality habitat and could support abundant wild prey populations, ..."

Please also note our previous revisions that discussed the limitations of NDVI in lines 451-454 and the need for better measures of site-level prey availability in lines 295-296:

-Lines 451-454: "Though the correlation between primary production and wild prey availability may be decoupled in protected areas experiencing large-scale defaunation in recent decades⁸⁸, we chose to include NDVI in our study as the closest proxy available for broad-scale patterns in site-level resource availability, including wild prey."

-Lines 295-296: "Site-specific estimates of wild prey availability could similarly improve our ability to explain lion responses to human disturbance, as the link between primary productivity and wild prey populations can be decoupled by management strategies or poaching pressures^{88,89}."